# Integrated rapid risk assessment for dengue fever in settings with limited diagnostic capacity and uncertain exposure: Development of a methodological framework for Tanzania

Matthias Hans Belau[1]*[☊], Juliane Boenecke[2☊], Jonathan Ströbele[2], Mirko Himmel[3], Daria Dretvić[3], Ummul-Khair Mustafa[4], Katharina Sophia Kreppel[4,5], Elingarami Sauli[4], Johanna Brinkel[2], Ulfia Annette Clemen[6], Thomas Clemen[6], Wolfgang Streit[3], Jürgen May[2‡], Amena Almes Ahmad[7‡], Ralf Reintjes[7‡], Heiko Becher[8‡]

1 University Medical Centre Hamburg-Eppendorf, Institute of Medical Biometry and Epidemiology, Hamburg, Germany, 2 Department of Infectious Disease Epidemiology, Bernhard Nocht Institute for Tropical Medicine, Hamburg, Germany, 3 Department for Microbiology and Biotechnology, University of Hamburg, Institute for Plant Sciences and Microbiology, Hamburg, Germany, 4 The Nelson Mandela African Institution of Sciences and Technology, School of Life Sciences and Bioengineering, Arusha, Tanzania, 5 Department of Public Health, Institute of Tropical Medicine, Antwerp, Belgium, 6 Department Computer Sciences, Hamburg University of Applied Sciences, Hamburg, Germany, 7 Department Health Sciences, Hamburg University of Applied Sciences, Hamburg, Germany, 8 Heidelberg University Hospital, Heidelberg Institute of Global Health, Heidelberg, Germany

☊ Shared first authorship (these two authors contributed equally to this work)
‡ Shared last authorship
* m.belau@uke.de

## Abstract

### Background

Dengue fever is one of the world's most important re-emerging but neglected infectious diseases. We aimed to develop and evaluate an integrated risk assessment framework to enhance early detection and risk assessment of potential dengue outbreaks in settings with limited routine surveillance and diagnostic capacity.

### Methods

Our risk assessment framework utilizes the combination of various methodological components: We first focused on (I) identifying relevant clinical signals based on a case definition for suspected dengue, (II) refining the signal for potential dengue diagnosis using contextual data, and (III) determining the public health risk associated with a verified dengue signal across various hazard, exposure, and contextual indicators. We then evaluated our framework using (i) historical clinical signals with syndromic and laboratory-confirmed disease information derived from WHO's Epidemic Intelligence from Open Sources (EIOS) technology using decision tree analyses, and (ii) historical dengue outbreak data from Tanzania at the regional level from 2019 (6,795 confirmed cases) using negative binomial regression analyses adjusted for month and region. Finally, we evaluated a test signal

**Data availability statement:** All relevant data are publicly available [from https://github.com/MARS-Group-HAW/esida-db] and within the manuscript and its supporting files.

**Funding:** This study was supported by the Epidemiologic Surveillance for Infectious Diseases in Sub-Saharan Africa (ESIDA) project. The ESIDA project itself was funded by the German Federal Ministry of Education and Research (BMBF - https://www.bmbf.de) under grant number 01DU20005(A-D) to JM. MHB received financial support from ESIDA grant number 01DU2005C; JuB from grant numbers 01DU2005A and 01DU2005B; MH and DD from grant number 01DU2005D; UKM, JoB, and KSK from grant number 01DU2005A; JS and UAC from grant number 01DU2005B. The funders had no role in study design, data collection and analysis, decision to publish, or preparation of the manuscript.

**Competing interests:** The authors have declared that no competing interests exist.

across all steps of our integrated framework to demonstrate the implementation of our multi-method approach.

## Results

The result of the suspected case refinement algorithm for clinically defined syndromic cases was consistent with the laboratory-confirmed diagnosis (dengue yes or no). Regression between confirmed dengue fever cases in 2019 as the dependent variable and a site-specific public health risk score as the independent variable showed strong evidence of an increase in dengue fever cases with higher site-specific risk (rate ratio = 2.51 (95% CI = [1.76, 3.58])).

## Conclusions

The framework can be used to rapidly determine the public health risk of dengue outbreaks, which is useful for planning and prioritizing interventions or for epidemic preparedness. It further allows for flexibility in its adaptation to target diseases and geographical contexts.

### Author summary

In our study, we set out to address the challenges posed by re-emerging dengue fever, a significant infectious disease globally. The goal was to create a comprehensive risk assessment framework that enhances the early detection and evaluation of potential dengue outbreaks, especially in regions with limited surveillance and diagnostic capabilities. By combining clinical signals, contextual data, and public health risk indicators, we developed an integrated approach to assess the likelihood and impact of dengue outbreaks. Our evaluation involved analyzing real-world data from the World Health Organization and Tanzania. The results demonstrated the effectiveness of our framework in accurately predicting the risk of dengue outbreaks. Specifically, our disease classification algorithm successfully matched suspected cases (outbreak signals) with laboratory-confirmed diagnoses, and we found a strong association between site-specific public health risk and the incidence of dengue fever cases. These findings are crucial as they provide a practical tool for predicting and preparing for dengue outbreaks. By improving early detection and risk assessment, our framework can help stakeholders plan and implement interventions more effectively, ultimately contributing to better public health outcomes. This research highlights the importance of interdisciplinary approaches in tackling infectious diseases and underscores the significance of proactive measures in controlling outbreaks.

## Introduction

Mosquito-borne dengue fever has become an alarming global public health threat, affecting 50% of the world's population [1]. Infection is caused by dengue virus (DENV), a flavivirus with four distinct serotypes (DENV 1-4) [2], and can lead to severe disease in the form of dengue hemorrhagic fever (DHF) and dengue shock syndrome (DSS) [3]. Global prevalence coincides with the geographic distribution of

its primary vector, urban *Aedes* mosquitoes, which are primarily found in tropical and subtropical regions [4, 5]. Both are expanding rapidly due to global warming, urban development, and land-use changes [6], highlighting the need for integrated preparedness, prevention and response [1].

Dengue outbreaks are predominantly reported in the Americas, South-East Asia, and the Western Pacific regions. Incidences have also been documented across Africa, with frequent outbreaks mainly occurring in East African countries. However, the true prevalence of dengue is likely to exceed the reported figures due to underreporting and misdiagnosis [7]. Given this context, and the lack of effective vaccines and antiviral treatments [8], early outbreak detection is critical to prevent serious cases and deaths, and mitigate social and economic harm from dengue epidemics [9–11].

When routine surveillance and diagnostic capacities are limited, employing event-based surveillance (EBS) is an effective strategy for early outbreak detection [12]. EBS can include reports on syndromes sensitive to dengue [13], or informal signals suggestive of dengue events, e.g., from news media or social networks, among others [14]. Sources that consolidate this type of intelligence are, for instance, the WHO's Epidemic Intelligence from Open Sources (EIOS) technology [15], as well as the web-based platforms ProMED [16], and HealthMap [17]. However, such systems may face limitations due to their high sensitivity to outbreak events, particularly arboviral diseases, requiring substantial screening efforts to verify signals. When combined with disease-specific geographic, demographic, and other risk indicators, signals could be refined, thereby enhancing EBS effectiveness.

A few conceptual frameworks [18], mobile software applications [19], and Early Warning, Response and Alert System (EWARS) tools exist for integrated dengue outbreak detection and risk assessment. Most of these resources are specifically designed for dengue-endemic settings, including Brazil [20], and Mexico [21, 22]. For example, disease trends are illustrated over time at district level by depicting the number of suspected cases within the expected range, determined by historical mean and standard deviation. Any number surpassing this average is considered suggestive of an unusual event [20, 21]. In addition, contextual alert indicators are monitored, such as meteorological (e.g., ambient temperature, relative humidity, and accumulated rainfall) and entomological factors (e.g., *Aedes* infestation rates using mosquito traps). However, many of these tools are not easily transferable to country contexts as they require routine surveillance and timely diagnostics as well as validated alarm indicators, all of which are often lacking in less-studied and resource-restricted settings like Sub-Saharan Africa [9,23]. Moreover, existing risk assessments in the African region, such as in response to the escalating dengue outbreaks in 2023, have been concentrated at national level [24], neglecting differing vulnerabilities and response needs within the individual countries.

Focusing on Tanzania in East Africa, we developed and evaluated an integrated risk assessment framework aimed at enhancing the early detection and risk determination of potential dengue outbreaks in settings with limited routine surveillance and diagnostic capacity. Our approach begins by identifying signals of suspected dengue fever cases through EBS, followed by a novel disease- and site-specific risk evaluation encompassing selected hazard, exposure, and contextual indicators. Our framework serves three key goals: first, to enhance the interpretation of suspected dengue case signals through ecological context when laboratory confirmation is unavailable; second, to evaluate factors influencing potential outbreak dynamics and impact at relevant spatial scales; and third, to generate public health risk profiles through integrated assessment. We tested this framework for conceptual feasibility using real-world data from WHO and Tanzania.

## Methods

### Ethics statement

The work presented is an exploratory study. The research was approved by the Ethics Committee of the Hamburg State Medical Chamber in Germany (ref. no. 2022-300221-WF) as well as Tanzania Northern Zone Health Research Ethics Committee (ref. no. KNCHREC00061/12/2021, issued: 31st January 2022).

### Test site Tanzania

The United Republic of Tanzania (Mainland and Zanzibar), located in East Africa, is divided into 31 regions and 195 districts, with a total population of 61.7 million (rural: 40.2 million, urban: 21.5 million) and an area of 945,087km² [25]. The climate is mainly tropical, with regional variations (e.g., humid and hot at the tropical coast, cooler and arid in the central plateau), and seasonal weather dynamics (heavy rains from mid-March to May, short rainy period from November to mid-January, dry period from May to October) [26].

The country has experienced recurrent dengue outbreaks of smaller and larger extent since 2010 [27–30]. A small outbreak in 2011 has been described by a single-center study that recruited 60 suspected cases seen at a private clinic in Dar es Salaam region between April and July, during rainy season, of which 45 cases were laboratory confirmed [31]. In February 2014, the Ministry of Health reported a major outbreak that had spread to seven regions in mainland Tanzania and two regions on the island of Zanzibar. By the end of May, there were 1,017 confirmed cases on the mainland, with 99% of the mainland cases reported from Dar es Salaam [32]. In March 2018, 11 dengue cases were reported in the coastal cities Dar es Salaam and Tanga [30]. The latest reported dengue outbreak so far, recorded in 2019, saw 6,795 confirmed cases. This represents the most comprehensively documented outbreak, with monthly data available at the regional level [33, 34]. In addition, Tanzania faces the emergence of various infectious diseases while limited in diagnostic, routine surveillance, and vector control capacity [27], amplifying its susceptibility to misdiagnosis, underreporting, and potential dengue epidemics.

### Integrated rapid risk assessment framework

Our rapid risk assessment framework adopts a multimethod approach, consisting of three compartments: (I) identifying relevant clinical signals based on a case definition for suspected dengue, (II) refining the signal for potential dengue diagnosis using contextual data, and (III) determining the public health risk associated with a verified dengue signal across various hazard, exposure, and contextual indicators. The framework and each methodological compartment are illustrated in Fig 1, and is described in detail in the subsequent sections.

**Compartment I: Signal identification according to the dengue case definition.** While most cases of dengue result in mild or no symptoms, in rare occasions, and often due to insufficient clinical evaluation, it can lead to potentially fatal complications [35]. Symptomatic dengue infection is characterized by fever and non-specific signs and symptoms, such as headache, vomiting, rash, myalgia or arthralgia, pain behind the eyes, positive tourniquet test, and low white blood cell count (dengue-like illness). In addition, warning signs can occur that point towards severe illness (DHF, DSS), which include severe abdominal pain, persistent vomiting, and mucosal bleeding [36].

Therefore, the entry point of our framework shall be clinical signals indicating an acute public health event based on the detection of *suspected dengue cases* (considering both, dengue-like illness and DHF), independent of any epidemiological link. Our case definition is detailed in Table 1, which aligns with international guidelines [14,24,29,37,38].

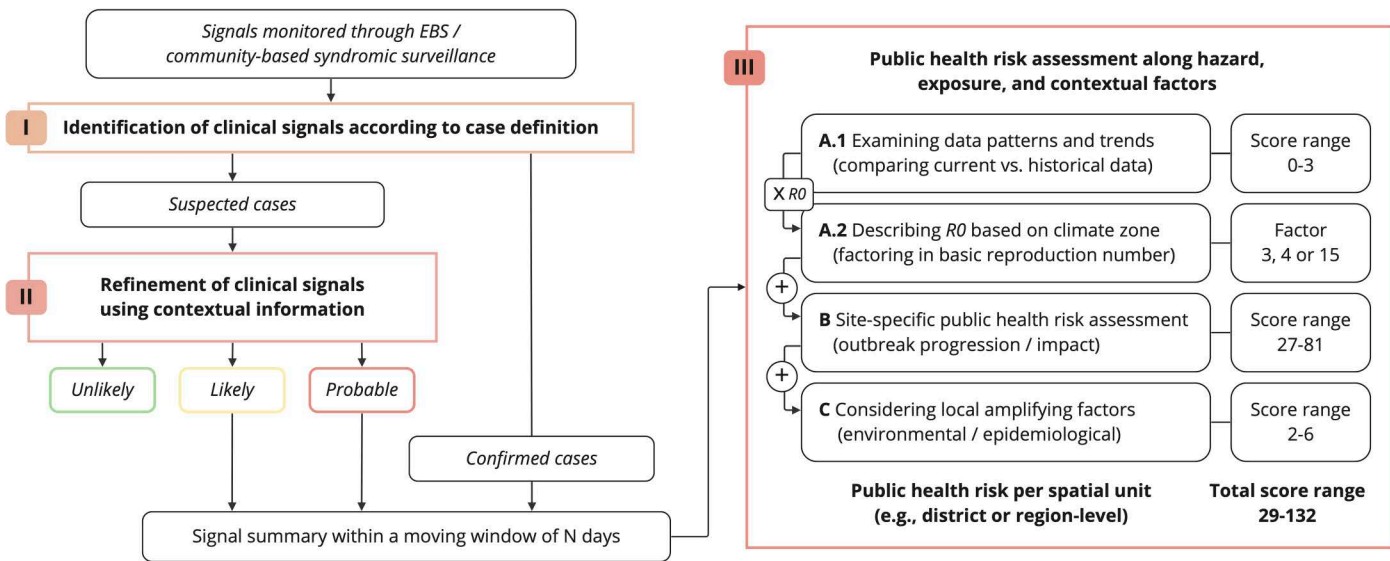

**Fig 1. Integrated rapid risk assessment framework following a multimethod approach.**

**Table 1. Case definition for dengue fever according to international guidelines.**

| Signal | Case Definition |
|---|---|
| *Suspected case* | **Dengue-sensitive syndrome WITHOUT warning signs (indicating mild to moderate disease):**<br>Any person who presents acute fever (≥37.5°C), usually from 2-7 days duration, AND at least two of the following manifestations – nausea/vomiting; abdominal pain; chills; rash; conjunctivitis; headache/retro-orbital pain; myalgia and arthralgia; petechial or positive tourniquet test (+ >10 pinpoint-sized spots of bleeding under the skin (petechiae) per square inch); low platelet count (thrombocytopenia); low white blood cell count (leukopenia) even without any warning sign or sign of severity |
| | **Dengue-sensitive syndrome WITH warning signs (indicating severe disease incl. DHF):**<br>Any person who presents acute fever (≥37.5°C)/history of fever in the past 24hours AND at least one of the following symptoms – intense abdominal pain (continuous or sustained); persisting vomiting and/or diarrhea; severe headache; mucosal bleeding; sensory disorder; feeling of faintness; fluid accumulation; respiratory distress; liver enlargement; signs of hypovolemic shock and rapid decline in platelet count with an increase in hematocrit |
| *Confirmed case* | A suspected case AND recent infection confirmed by rapid diagnostic tests (preferred laboratory tool for diagnosing dengue due to resource constraints in the country/highly suggested laboratory confirmation) |

Signals relevant to our framework shall be gathered from EBS or, when possible, identified through community-based syndromic surveillance. Additionally, confirmed cases (see Table 1, e.g., registered through routine surveillance or reported from sentinel sites), as well as the supplementation of epidemiological links (e.g., recent travel to dengue-endemic or outbreak-affected areas, recent contact to infected individuals [36]) can support signal verification, provided that diagnostic and outbreak investigation capacities are available in a timely manner.

**Compartment II: Refinement of clinical signals using contextual data.** Our framework targets settings with constrained resources, where timely epidemiological or laboratory verification of signals may not be available. Therefore, we incorporated contextual factors that enable the refinement of signals through the ecological understanding of dengue, a vector-borne

disease primarily transmitted by *Aedes aegypti* mosquitoes, until verification can be achieved through laboratory confirmation. Our refinement algorithm comprises five context-driven indicators, categorized into *Survival*, *Suitability*, and *Plausibility Conditions*, each informed by real-time data (see Fig 2).

On average, *Aedes aegypti*'s life cycle from egg to adult takes 8–10 days [39, 40], requiring aquatic environment. The lifespan of female adults varies between 10-35 days [41–43], while males typically live for 3–6 days [42, 43]. After taking an infected blood meal, female mosquitoes require 8–12 days extrinsic incubation period (EIP) before becoming infectious [44]. In humans, symptoms then typically appear 4–7 days post-infection [44].. Given these biological parameters in both vector and human host, we apply a retrospective period of 14 days after symptomatic case detection (day 0) as a practical compromise for rapid assessment, accounting for human symptom development (set as day -4), viral EIP (set as day -14), assuming a stable mosquito population with transmission capacity during prolonged rainy conditions. To further specify environmental conditions, we identified temperature, precipitation and humidity as key factors in *Aedes aegypti* mosquitoes, which impact all life stages (eggs, larvae, pupae, adult), including reproduction, behavior and survival [45–52]. However, while the effects of temperature are well-understood, the relationships between precipitation and humidity (i.e., quantity and frequency) remain underexplored. Given this context and historical data from Tanzania that indicate dengue outbreaks are more prevalent in the rainy season [53–55], we have incorporated seasonality (dry vs. rainy) into our framework to account for variations in precipitation and humidity. A more detailed description of the *Survival* and *Suitability Conditions* can be found in S1 Text.

Besides temperature and seasonality, a predominantly urban environment (>10% urban landscape [56]) has been positively associated with *Aedes aegypti* abundance and dengue transmission [57–59], given the mosquitoes' feeding (primarily on humans) and breeding preferences (proximity to blood sources in densely populated areas) [52]. Therefore, we included the urbanization level as well as historical outbreak data to verify the plausibility of conducive environmental factors [29,60–62]. Given the absence of routine dengue surveillance, we consider case reports within the 2010–2023 reference period independently of the timing (e.g., calendar week or month) of signal occurrence.

As illustrated in Fig 2, refinement of clinical signals is carried out based on the contextual indicators described: Signals are first clustered in space and time and then classified according to their potential cause using the dengue-sensitive algorithm. As a result, a signal shall be classified as "Unlikely" (level 1), "Likely" (level 2 and 3) or "Probable" (level 4 and 5), allowing for quick risk evaluation while laboratory confirmation is pending.

**Compartment III: Site-specific risk assessment for dengue.**  In outbreak situations, swiftly evaluating the associated public health risk is crucial to inform adequate response. Therefore, a rapid risk assessment is employed to generate an up-to-date risk profile for the affected area, as recommended by WHO [63, 64] or the European Centre For Disease Prevention and Control (ECDC) [65]. This procedure informed Compartment III:

Our site-specific risk assessment, tied to a refined or verified signal, evaluates three key elements: (i) the hazard posed by mosquito-borne dengue, (ii) potential exposure to the hazard, and (iii) the context in which the hazard occurs. Dengue hazard shall be quantified by its basic reproductive number (R0), defined as the number of secondary infections produced by the index case, which averages R0 >3 in temperate zones, R0 >4 in tropical zones, and R0 >15 in subtropical zones [66]. Hazard exposure shall be determined by the number of individuals or groups known or likely to be exposed or susceptible to dengue infection. This may be derived from immunity levels in the population as well as the prevalence of the mosquito vector. However, estimating immunological factors is complex due to the diversity of serotypes and

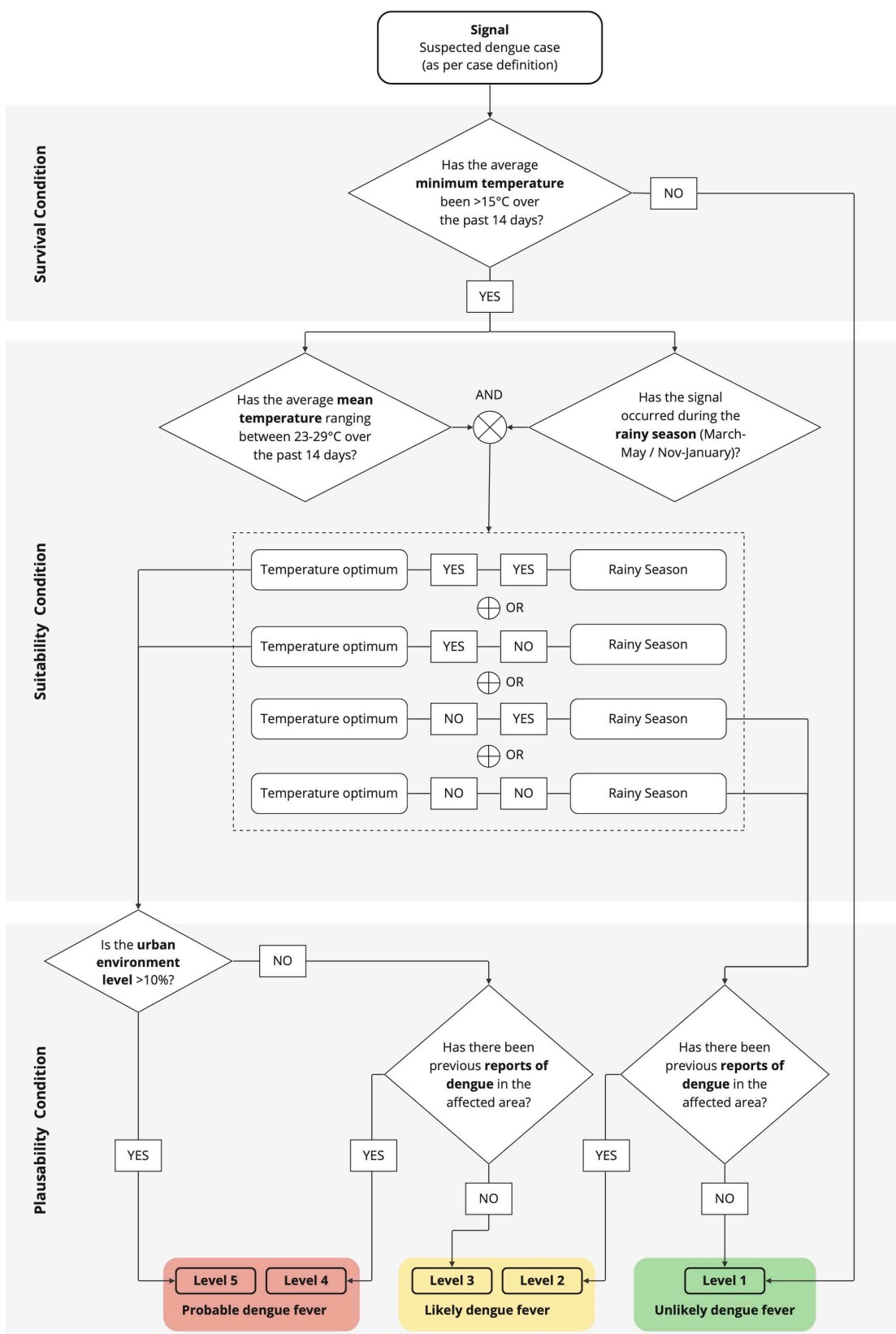

**Fig 2. Signal refinement algorithm using contextual indicators.**

underreporting in African countries [7]. Conversely, the abundance of the carrier mosquito and/or favorable ecological conditions are well-documented and often utilized as exposure indicators [67]. Lastly, contextual assessment of the hazard shall include socioeconomic, environmental, and programmatic factors.

Upon reviewing scientific literature and policy documents, we identified several factors across different categories (diagnostic, surveillance, and healthcare capacity; demographic and socioeconomic factors; infrastructure; knowledge, attitudes, and behavior; ecological factors) associated with the public health risk of dengue outbreaks. Following expert discussions among epidemiologists, tropical medicine and public health physicians and scientists, microbiologists, and entomologists, these factors were first ranked in order of importance and then classified as either generic or dengue-specific. We further clustered these factors according to their potential influence on (i) outbreak progression (Table 2A, 21 indicators) and (ii) their likely impact on the population and economy (Table 2B, 7 indicators). S2 Text provides a thorough explanation of the rationale behind selecting a total of 28 indicators.

To determine the local risk profile, we assign scores ranging from one to three to each indicator, where 1 indicates low risk, 2 indicates medium risk, and 3 signifies high risk. The primary data sources informing our scoring approach (i.e., indicator cut-offs) were the World Bank [69], WHO [74], World Economic Forum [73], European Union Copernicus Earth Observation Program [56], and Malaria Atlas Project [72]. The method for determining an indicator cut-off is based on tertile classification, using openly available data for all 193 United Nations member countries for each risk indicator listed in Table 2. For indicators requiring qualitative assessment due to insufficient data, we applied a scoring system using minimum and maximum values, where a score of one indicates 'yes' and three indicates 'no', or vice versa (e.g., proof of vector abundance).

## Determining a public health risk profile using the integrated rapid risk assessment framework

To assess the public health risk associated with a potential dengue outbreak event, we first gather all clinical signals matching our case definition (*Compartment I*) within a defined time period (e.g., moving window of last 7 days) and spatial unit (e.g., region, district, or subdistrict level). All signals sensitive to dengue-like illness or DHF (see case definition Table 1) will be refined through *Compartment II* of our framework, pending laboratory confirmation, and divided into the classes "Unlikely", "Likely", and "Probable". In addition, confirmed cases (see Table 1) may be collected. All signals are then systematically documented, following the template provided in Table 3.

The first step in quantifying the public health risk associated with the collected signals (*Compartment III*) involves assigning a *Risk Score A* by examining the data trends, and factoring in the climate-informed *R0*. This is done by comparing all signals gathered during the defined period against historical averages from the reference period, typically calculated from recent years (e.g., average of the last 3 to 5 years). A potential public health concern emerges when the number of current signals surpasses the number of historical signals (N[current signals]-N[historical signals]>0). Accordingly, each signal category relevant to our dengue framework (i.e., laboratory confirmed signals and those classified "Probable" and "Likely") receives a score of 1 if the difference is greater than zero (see Table 3, i.e., a>e, b>f, c>g) and a score of 0 if the difference is zero or negative (see Table 3, i.e., a≤e, b≤f, c≤g). Following this approach, *Risk Score A* can range from a maximum of three relevant signal categories (N[current signals]-N[historical signals]>0 in all three categories), to a minimum of zero (N[current signals]-N[historical signals] ≤0 in all three categories), which is then multiplied

**Table 2.  A. Site-specific risk characterization for dengue fever outbreak with focus on likelihood to spread. B. Site-specific risk characterization for dengue fever outbreak with focus on population and economic impact.**

**(A)**

| Category/ Risk factor | Risk level | | | Threshold derived from |
|---|---|---|---|---|
| | 1 (low) | 2 (medium) | 3 (high) | |
| *Outbreak progression* | | | | |
| *Generic* | | | | |
| *Health care capacity and access* | | | | |
| Medical laboratory personnel density (per 10,000 population) | >2.6 | 1.1–2.6 | <1.1 | Information available for 135 countries; data reference year 2000–2021 [68] |
| *Sociodemographic* | | | | |
| Population density (population/km²) | <58.2 | 58.2–158.9 | >158.9 | Information available for 193 countries; data reference year 2021 [69] |
| Household members (average) | <3.0 | 3.0–4.3 | >4.3 | Information available for 173 countries; data reference year 2000–2021 [70] |
| *Socioeconomic* | | | | |
| Percentage of males achieving secondary education or higher† | >94.8 | 74.6–94.8 | <74.6 | Information available for 142 countries; data reference year 2019–2022 [69] |
| Percentage of females achieving secondary education or higher† | >97.3 | 78.0–97.3 | <78.0 | Information available for 138 countries; data reference year 2018–2022 [69] |
| Gini index (income inequality index) (0(low) to 1(high)) | <0.3 | 0.3–0.4 | >0.4 | Information available for 96 countries; data reference year 2018–2022 [69] |
| Percentage of population with less than 2.15 USD per day (poverty gap) | <0.4 | 0.4–2.8 | >2.8 | Information available for 97 countries; data reference year 2018–2022 [69] |
| *Dengue–specific* | | | | |
| *Health care capacity and access* | | | | |
| Rapid diagnostic test (RDT) availability (in the country) | Yes | | No/Unknown | WHO recommendation [71] |
| Confirmatory diagnostics, e.g., reference lab (in the country) | Yes | | No/Unknown | WHO recommendation [71] |
| *Sociodemographic* | | | | |
| Percentage of population aged 15–49 years (mobile/ working population) | <47.6 | 47.6–51.1 | >51.1 | Information available for 193 countries; data reference year 2022 [72] |
| *Mobility infrastructure* | | | | |
| Quality of airport infrastructure (1(low) to 7(high)) | <4.1 | 4.1–5.0 | >5.0 | Information available for 140 countries; Data reference year 2019 [73] |
| Quality of seaport infrastructure (1(low) to 7(high)) | <3.5 | 3.5–4.5 | >4.5 | Information available for 140 countries; Data reference year 2019 [73] |
| Quality of railroad infrastructure (1(low) to 7(high)) | <2.7 | 2.7–3.9 | >3.9 | Information available for 140 countries; Data reference year 2019 [73] |
| Quality of road infrastructure (1(low) to 7(high)) | <3.5 | 3.5–4.5 | >4.5 | Information available for 140 countries; Data reference year 2019 [73] |
| *Water, sanitation, and hygiene* | | | | |
| Percentage of population with access to at least basic water infrastructure | >94.9 | 57.8–94.9 | <57.8 | Information available for 123 countries; data reference year 2022 [69] |
| Percentage of population with access to at least basic sanitation | >82.5 | 41.6–82.5 | <41.6 | Information available for 127 countries; data reference year 2022 [69] |
| Percentage of population with access to at least basic garbage collection | >92.3 | 67.0–92.3 | <67.0 | Information available for 82 countries; data reference year 2010–2020 [69] |
| *Knowledge, attitudes, and behavior* | | | | |
| Percentage of population using at least one Insecticide-treated Net (ITN) | >63.7 | 52.0–63.7 | <52.0 | Information available for 40 countries; data reference year 2021 [74] |
| Proof of vector abundance (in the country) | No | | Yes/Unknown | WHO recommendation [71] |

*(Continued)*

**Table 2.** (Continued)

**(A)**

| Category/ Risk factor | Risk level | | | Threshold derived from |
|---|---|---|---|---|
| | **1 (low)** | **2 (medium)** | **3 (high)** | |
| Vector control strategies available (in the country) | Yes | | No/Unknown | WHO recommendation [71] |
| *Ecology and environment* | | | | |
| Proportion of urban environment/ built-up index (0(low) to 1(high))‡ | <0.1 | 0.1–0.2 | >0.2 | According to the classes of the degree of urbanization [56] |

**(B)**

| Category/ Risk factor | Risk level | | | Threshold derived from |
|---|---|---|---|---|
| | **1 (low)** | **2 (medium)** | **3 (high)** | |
| *Outbreak impact on the population and economy* | | | | |
| *Generic* | | | | |
| *Health care capacity and access* | | | | |
| Physician density (per 10,000 population) | >26.1 | 7.3–26.1 | <7.3 | Information available for 193 countries; data reference year 2002–2021 [74] |
| Nurses and midwife density (per 10,000 population) | >56.0 | 19.8–56.0 | <19.8 | Information available for 193 countries; data reference year 2014–2021 [74] |
| Hospital beds (per 10,000 population) | >30.0 | 14.0–30.0 | <14.0 | Information available for 178 countries; data reference year 2004–2019 [74] |
| Healthcare accessibility (motorized travel time to nearest health facility in minutes)# | <33.4 | 33.4–136.5 | >136.5 | Information available for 193 countries; data reference year 2022 [72] |
| *Dengue-specific* | | | | |
| *Sociodemographic* | | | | |
| Percentage of population aged 5–39 years (morbidity relevant ages) | <47.6 | 47.6–62.4 | >62.4 | Information available for 193 countries; data reference year 2022 [69] |
| Percentage of population aged <15 years (mortality relevant ages) | <19.1 | 19.1–30.3 | >30.3 | Information available for 193 countries; data reference year 2022 [69] |
| Percentage of population aged 65+ years (mortality relevant ages) | <4.9 | 4.9–11.9 | >11.9 | Information available for 193 countries; data reference year 2002–2021 [69] |

†*Half the total weight for secondary education or higher as a risk factor;*

‡*Rural = 0.13 to suburban = 0.21 to urban = 0.23.*

#The following steps were taken: First, the GeoTIFF raster file provided by the Malaria Atlas Project [75] was processed against the shape file of the 193 countries considered. The raster cells within each country geometry were isolated using Python library Rasterio and then further processed using NumPy [76] to calculate the average travel time within the country based on all cell values.

**Table 3. Signal input, processing, and output by signal category.**

| | Signal category | | | |
|---|---|---|---|---|
| | **Laboratory-confirmed dengue cases**† | **Refined dengue-signal** | | |
| | | **Probable cases** | **Likely cases** | **Unlikely cases**ƒ |
| *Current* | a | b | c | d |
| *Historical* | e | f | g | h |
| *Potential of concern*‡ | a>e | b>f | c>g | d>h |

†Confirmed dengue case by rapid diagnostic test (antibody or antigen) and/or polymerase chain reaction (PCR);

‡Number above historical average/expected level; ƒ Febrile illness with other manifestations/symptoms may indicate other hemorrhagic fever.

by the dengue hazard, expressed as *R0*, specific to the climate zone (multiplied by 3, 4, and 15 for temperate, tropical, and subtropical zones [66], respectively). This first step of our risk assessment framework assesses the rate at which dengue can propagate through a population, where a higher value indicates a greater outbreak risk. Table 4 summarizes the corresponding signal-based risk score, taking into account historical comparison and *R0* associated with different climates.

In a next step, the signal-based *Risk Score A* (ranging from 0-45) is augmented by incorporating our site-specific risk assessment, *Risk Score B*, which takes into account contextual factors of the corresponding spatial unit (see Table 2). From the site-specific assessment across two dimensions (outbreak progression and impact), scores can range from a minimum of 27 to a maximum of 81, resulting in a total risk score variation (*Risk Score A+B*) between 27 and 126 per spatial unit. Again, a higher score indicates a greater potential for an outbreak of significant public health concern.

In a final step, we account for amplifying factors, *Risk Score C*, by including data on seasonality and recent local outbreak events (confirmed cases reported within the country and in adjacent regions of neighboring countries). Considering Tanzania's three seasons, a signal during the dry season corresponds to a low risk level (score 1), the shorter rainy season to a medium risk level (score 2), and the longer rainy season, which experiences the heaviest rainfall, to a high-risk level (score 3). Similarly, no or at least one laboratory confirmed dengue case detected within the last four weeks in the country but not in an adjacent region, corresponds to a low risk level (score 1), at least one case in the same or adjacent region (of a neighboring country) but not in the same district corresponds to a medium risk level (score 2), and at least one case in the same district or subdistrict corresponds to a high risk level (score 3). Thus, the final risk score (*Risk Score A+B+C*) for each spatial unit can range from 29-132.

## Framework validation

We validated our framework in two stages, following the WHO Guideline on Monitoring and Evaluating Digital Health Interventions (conceptual stage) [77]. First, to demonstrate the feasibility of our signal refinement algorithm for suspected dengue (*Compartment II*), we reviewed the WHO AFRO Weekly Bulletins [78] and WHO EIOS system [15] for signals in the period 01/2019-12/2023 that align with our case definition (*Compartment I*). We identified three signals for which we found comprehensive descriptions, including confirmation through laboratory diagnostics. These occurrences were noted in March 2023 [79, 80], July 2022 [81,82], and January 2019 [83], respectively. A description of the test signals is given in Table 5. Test signals were then classified through the algorithm (*Compartment II*, see Fig 2) and the outcome compared to the actual diagnoses.

**Table 4.  Signal-based dengue outbreak risk score by climate zone.**

| # of relevant signal categories[†] of potential of concern[‡] | Temperate zone (# multiplied by 3) | Tropical zone (# multiplied by 4) | Subtropical zone (# multiplied by 15) |
|---|---|---|---|
| *0* | 0 | 0 | 0 |
| *1* | 3 | 4 | 15 |
| *2* | 6 | 8 | 30 |
| *3* | 9 | 12 | 45 |

# Number,

[†]Laboratory-confirmed signals or signals classified as probable dengue fever,

[‡]Number above historical average/expected level (see Table 3).

**Table 5. Test signals based on case reports of suspected dengue for preliminary signal verification.**

|  | Test signal #1 | Test signal #2 | Test signal #3 |
|---|---|---|---|
| *Signal data* |  |  |  |
| *Report* | 2023-03-16 | 2022-07-05 | 2019-01-31 |
| *Location* | Zone: Lake zone, tropical climate | Zone: Southern zone, tropical climate | Zone: Eastern zone, tropical climate |
|  | Region: Kagera | Region: Lindi | Region: Dar es Salaam and Tanga |
|  | District: Bukoba Vijijini (Bukoba DC) | District: Ruangwa | District: Unknown |
|  | Wards: Maruku, Kanyangereko | Wards: Unknown | Wards: Unknown |
| *Signal* | Unclassified cases of febrile illness with bleeding signs | Unclassified cases of febrile illness with bleeding signs | Suspected dengue fever cases |
| *Description* | Seven individuals are showing fever, bleeding in various parts of the body, and kidney failure. Five of these individuals have passed away, while two are currently receiving hospital treatment. Progression of the illness indicates it might be infectious. | On July 5 and 7, two cases presented at a health center in Ruangwa District, Lindi Region, with symptoms such as fever, nosebleeds, headaches, and fatigue. | Since August 2018, Dar es Salaam and Tanga regions have each reported 19 suspected dengue cases, totaling 38. January 2019 saw the peak of reported cases. Out of these, 27 cases have been confirmed via the dengue rapid diagnostic test. |

In a second step, we evaluated the site-specific risk assessment framework (*Compartment III*, see Fig 2) using historical dengue case reports, clustered by month and region, from the outbreak year 2019. Assigning a risk score for each indicator of the framework was determined by comparing the reference values (2019) to the cut-off values specified in Table 2. Statistical analyses were then performed to validate the risk scores assigned to each district.

Finally, to showcase the implementation of our multimethod approach, we assessed a test signal across all steps of our integrated framework. A cumulative risk score, which includes signal-based risk (*Risk Score A*), contextual factors (*Risk Score B*), and amplifying risk factors (*Risk Score C*), was calculated for each district and mapped for illustrative purposes.

## Data accessibility

To inform our risk assessment framework, we searched for openly available epidemiological data (including reports on dengue occurring between 2010 and 2019 as well as dengue outbreak figures from 2019, see S1 Table), as well as contextual data for Tanzania spanning the period 2010-2023, which were accessible at different spatial and temporal resolutions. A summary of the contextual data inputs, including data sources, spatiotemporal details, and data quality, is provided in S2 Table. Data was harmonized at district level, according to the country's health reporting system [84], using an open-source Data Fusion Engine developed for public health data integration [85]. In cases where data was unavailable at district level, data from higher administrative levels (such as regional or national) were utilized instead.

## Statistical analyses

For statistical analyses, the site-specific risk scores were categorized into quintiles at the district level, with quintile 1 (Q1) representing the lowest and quintile 5 (Q5) representing the highest site-specific risk. We first performed descriptive analyses and created a quintile-based choropleth map to illustrate the regional distribution of site-specific risk (*Risk Score B*). We then estimated incidence rate ratios (IRR) for confirmed dengue fever cases in 2019 to explore the association with the site-specific risk score (model 1: excluding amplifying factors (*Risk Score B*); model 2: including amplifying factors (*Risk Score B and C*)) by fitting negative binomial regression models, with the log of the region's population count as the offset variable. The selection of both models aimed to understand the significance of temporally dynamic

factors (i.e., seasonality and case reports). Adjustments were made for month and area (North, East, South, Southern Highlands, Southwestern Highlands, Western, Northwestern, Zanzibar). We calculated 95% confidence intervals (CIs), and significance was set at p<0.05. All analyses were performed using STATA MP version 18.

## Results

In the following section, we detail the implementation and evaluation of our framework in three stages: (i) assessing historical signals in comparison to actual diagnoses using *Compartment I*, (ii) validating the site-specific risk scores for Tanzanian districts (*Risk Score B* only), independent of any signal, and (iii) assessing a historical signal throughout all compartments of our framework to demonstrate its application and feasibility.

(i)Validating the refinement of test signals derived from WHO AFRO and WHO EIOS reports

Table 6 shows the results of the test signals' refinement through our dengue-sensitive algorithm (*Compartment II*), along with the reference values used for signal classification.

Test signal #1 and test signal #2, due to their ecological profiles (particularly *Suitability Conditions*), were not distinctly categorized as "Probable" dengue and even yielded an "Unlikely" result without considering the *Plausibility Conditions*. The confirmed diagnoses were Marburg virus disease and leptospirosis, respectively, aligning with the results of our algorithm. However, given the favorable assessment of *Survival Conditions* and historical dengue reports in the regions of both test signals (*Plausibility Condition*), it is advisable to lean towards higher sensitivity (thus, "Likely" (level 2) instead of "Unlikely" (level 1)), prompting diagnostic clarification and clinical monitoring as a precautionary measure (i.e., differential diagnosis for other viral hemorrhagic fevers (VHF) and arboviral diseases relevant to Tanzania, see S3 Table).

In contrast, test signal #3 was distinctly classified as "Probable" dengue and met both the *Survival* and *Suitability Conditions*. Both *Plausibility Conditions* (urban environment and historical dengue reports) were also confirmed for test signal #3. Consequently, immediate confirmatory diagnostics and clinical monitoring of affected cases are strongly advised. Given the evidence pointing to dengue, conducting a real-time rapid risk assessment is also recommended to facilitate prompt outbreak response (e.g., risk awareness, outbreak investigation and vector control).

(ii) Validating the site-specific dengue risk assessment in Tanzania

According to our site-specific risk assessment (*Risk Score B*), the risk score for Tanzania varies from 48 (minimum) to 59 (maximum) across the 195 districts, with an arithmetic mean of 52.9 points. Longido DC and Ngorongoro DC in the Arusha region have the lowest site-specific risk, while Kondoa DC in the Songwe region and Mvomero DC in the Kagera region have the highest. Table 7 displays the site-specific risk scores aggregated by region, showing Arusha region as having the lowest risk, while both Geita and Tabora regions hold the highest site-specific risk level.

The distribution of site-specific risk quintiles is shown in Fig 3, illustrating significant spatial variation. Districts in the north-western regions seem particularly vulnerable to dengue outbreaks of major public health concern, whereas Tanzania's southeastern parts hold a lower site-specific risk. In the northeastern regions, districts with both higher and lower risk profile can be identified.

Adjusted negative binomial regression between confirmed dengue fever cases in 2019 as the dependent variable and the site-specific risk score excluding amplifying factors (*Risk Score B*) as the independent variable showed strong evidence of an increase in dengue fever cases

**Table 6. Refinement of suspected dengue signals across three test scenarios.**

|  | Test signal #1 | Test signal #2 | Test signal #3 |
|---|---|---|---|
| *Survival condition* | Has the minimum temperature been >15°C in the past 14 days (°C, mean, min-max)? | | |
| *Outcome* | Yes | Yes | Yes |
| *Reference value* | 17.42 (15.9-18.9) | 16.01 (14.8-17.3) | 25.76 (22-27) |
| *Suitability condition* | Has the mean temperature ranging between 23-29°C in the past 14 days (°C, mean, min-max)? | | |
| *Outcome* | No | No | Yes |
| *Reference value* | 21.69 (19.5-24) | 21.37 (20.3-22.4) | 28,55 (27,4-29,4) |
|  | Has the signal occurred during the rainy season (March-May, November-January)? | | |
| *Outcome* | Yes | No | Yes |
| *Reference value* | 03/2023 | 07/2022 | 01/2019 |
| *Plausibility condition* | Is the urban environment level >10% (proportion ranging from 0-1)? | | |
| *Outcome* | Not applicable | Not applicable | Yes |
| *Reference value* | -- | -- | 0.28746 |
|  | Has there been previous reports of dengue in the affected area (transmission hotspot since 2010)? | | |
| *Outcome* | Yes | Yes | Yes |
| *Reference value* | Outbreak 2019 | Outbreak 2019 | Outbreak 2010, 2013/14, 2018 |
| *Outcome w/o plausibility check* | Unlikely (level 1/4) | Unlikely (level 1/4) | Probable (level 5/5) |
| *Outcome w/ plausibility check* | Likely (level 2/4) | Likely (level 2/4) | Probable (level 5/5) |
| *Suggested action points* | Given the ecological context, the signal suggests a **potential case of dengue**. Verification through laboratory diagnostics and differential diagnosis from other febrile illnesses (especially hemorrhagic fevers) is strongly advised. It is recommended to evaluate the public health risk using the integrated framework as a precautionary measure. | Given the ecological context, the signal suggests a **potential case of dengue**. Verification through laboratory diagnostics and differential diagnosis from other febrile illnesses (especially hemorrhagic fevers) is strongly advised. It is recommended to evaluate the public health risk using the integrated framework as a precautionary measure. | Given the ecological context, the signal suggests a **probable case of dengue**. Immediate laboratory confirmation (differentiating from other viral hemorrhagic fevers and arboviruses) is necessary. Cases should be assessed for potential complications (DHF, DSS) and monitored accordingly. We strongly advise evaluating the public health risk using the integrated framework. |
| *Validation* | | | |
| *Confirmed diagnosis* | Marburg virus disease (PCR) | Leptospirosis (PCR) | Dengue virus disease (RDT) |
| *Report* | 2023-03-21 | 2022-07-18 | 2019-01-31 |

with higher site-specific risk scores (model 1: IRR = 2.51 (95% CI = [1.76, 3.58])). When the site-specific risk score including amplifying factors (*Risk Score B+C*) was used as the independent variable, adjusted negative binomial regression also showed strong evidence of an increase in dengue fever cases with higher site-specific risk scores (model 2: IRR = 2.12 (95% CI = [1.65, 2.72]).

(iii) Implementing the integrated public health risk assessment framework

Since only test signal #3 was classified as "Probable" dengue by our dengue-sensitive algorithm (*Compartment II*), and later confirmed by RDT, our showcase will focus on this particular event in the reference period January 25-31, 2019 (period of 7 days, including day of report on January 31, 2019). As outlined in Table 5, the signal identified 19 suspected cases in the Dar es Salaam and Tanga region (classified as "Probable" dengue), respectively, with 27 of these confirmed as dengue, though lacking spatial details (83). We therefore assume that confirmed cases have been reported in both regions, represented as "+/-27" in the following processing.

**Table 7. Site-specific risk score (mean, standard deviation (SD)) by region (including sub-scores *and total score*).**

| Region | Sub-score | | Total score (min, max: 27, 81) | Number of districts |
|---|---|---|---|---|
| | Progression (min, max: 20, 60) | Impact (min, max: 7, 21) | | |
| Arusha | 33.5 (2.5) | 17.0 (0.0) | 50.5 (2.5) | 7 |
| Dar es Salaam | 38.2 (1.3) | 16.4 (0.5) | 54.6 (1.3) | 5 |
| Dodoma | 34.7 (1.4) | 16.3 (0.5) | 51.1 (1.8) | 8 |
| Geita | 37.3 (1.2) | 18.0 (0.0) | 55.3 (1.2) | 6 |
| Iringa | 35.8 (2.1) | 16.8 (0.4) | 52.6 (2.4) | 5 |
| Kagera | 37.7 (1.9) | 16.6 (0.5) | 54.3 (2.1) | 8 |
| Katavi | 35.2 (1.6) | 18.0 (0.0) | 53.2 (1.6) | 5 |
| Kigoma | 35.5 (1.3) | 17.2 (0.4) | 52.7 (1.6) | 8 |
| Kilimanjaro | 36.7 (2.2) | 16.5 (0.5) | 53.2 (2.2) | 7 |
| Lindi | 35.0 (0.6) | 16.8 (0.4) | 51.8 (0.7) | 6 |
| Manyara | 36.0 (1.0) | 17.7 (0.4) | 53.7 (1.1) | 7 |
| Mara | 38.2 (0.8) | 16.6 (0.5) | 54.9 (1.0) | 9 |
| Mbeya | 36.8 (2.1) | 17.0 (0.0) | 53.8 (2.1) | 7 |
| Morogoro | 34.0 (2.1) | 16.7 (0.4) | 50.7 (2.2) | 9 |
| Mtwara | 35.6 (1.5) | 16.7 (0.4) | 52.4 (1.7) | 9 |
| Mwanza | 37.3 (2.2) | 15.6 (0.5) | 53.0 (2.5) | 8 |
| Njombe | 34.8 (1.3) | 16.0 (0.0) | 50.8 (1.3) | 6 |
| Pwani | 34.8 (1.2) | 16.7 (0.4) | 51.6 (1.3) | 9 |
| Rukwa | 35.0 (0.8) | 18.0 (0.0) | 53.0 (0.8) | 4 |
| Ruvuma | 34.5 (1.4) | 16.5 (0.5) | 51.0 (1.6) | 8 |
| Shinyanga | 37.5 (1.5) | 15.8 (0.4) | 53.3 (1.5) | 6 |
| Simiyu | 36.1 (0.9) | 18.0 (0.0) | 54.1 (0.9) | 6 |
| Singida | 36.2 (0.7) | 16.1 (0.3) | 52.4 (1.1) | 7 |
| Songwe | 36.8 (2.4) | 17.6 (0.5) | 54.4 (2.7) | 5 |
| Tabora | 38.3 (1.1) | 17.0 (0.0) | 55.3 (1.1) | 8 |
| Tanga | 37.6 (1.7) | 16.1 (0.4) | 53.8 (1.9) | 11 |
| Zanzibar | 36.0 (1.2) | 16.5 (0.5) | 52.5 (1.6) | 11 |
| Tanzania | 36.1 (2.0) | 16.7 (0.7) | 52.9 (2.1) | 195 |

The regional risk is determined by averaging the risk scores from each district within the region, referencing the administrative boundaries of Tanzania established in 2016.

Due to insufficient routine dengue surveillance in Tanzania, we lack adequate descriptions of cases from the last 3-5 years. Therefore, we estimate the historical baseline to be zero to compare our signal data against historical averages (see template Table 3). The assessment indicates that current cases surpass historical cases in the two categories "Probable" and "Confirmed" (i.e., $a>e$ (confirmed) and $b>f$ (probable"), where $a$ equals +/-27, $b$ equals 19, and $e$ as well as $f$ equal zero). As a result, each region is assigned two risk points, which are then multiplied by four ($R0$ for a tropical climate) as per framework *Risk Score A.1* and *A.2* (see Table 4). This calculation totals in a score of 8 out of a possible maximum of 45 for each region.

Next, the signal is processed through framework *Risk Score B* and *C*, focusing on the site-specific risk assessment at district level across 27 contextual and two amplifying indicators in the reference month January 2019. *Risk Score B* yields 54.6 points for the Dar es Salaam region (54-56 across all five districts) and 53.8 points for the Tanga region (52-58 across all 11 districts), each out of a maximum of 81 points. *Risk Score C* adds another four points to all districts of both

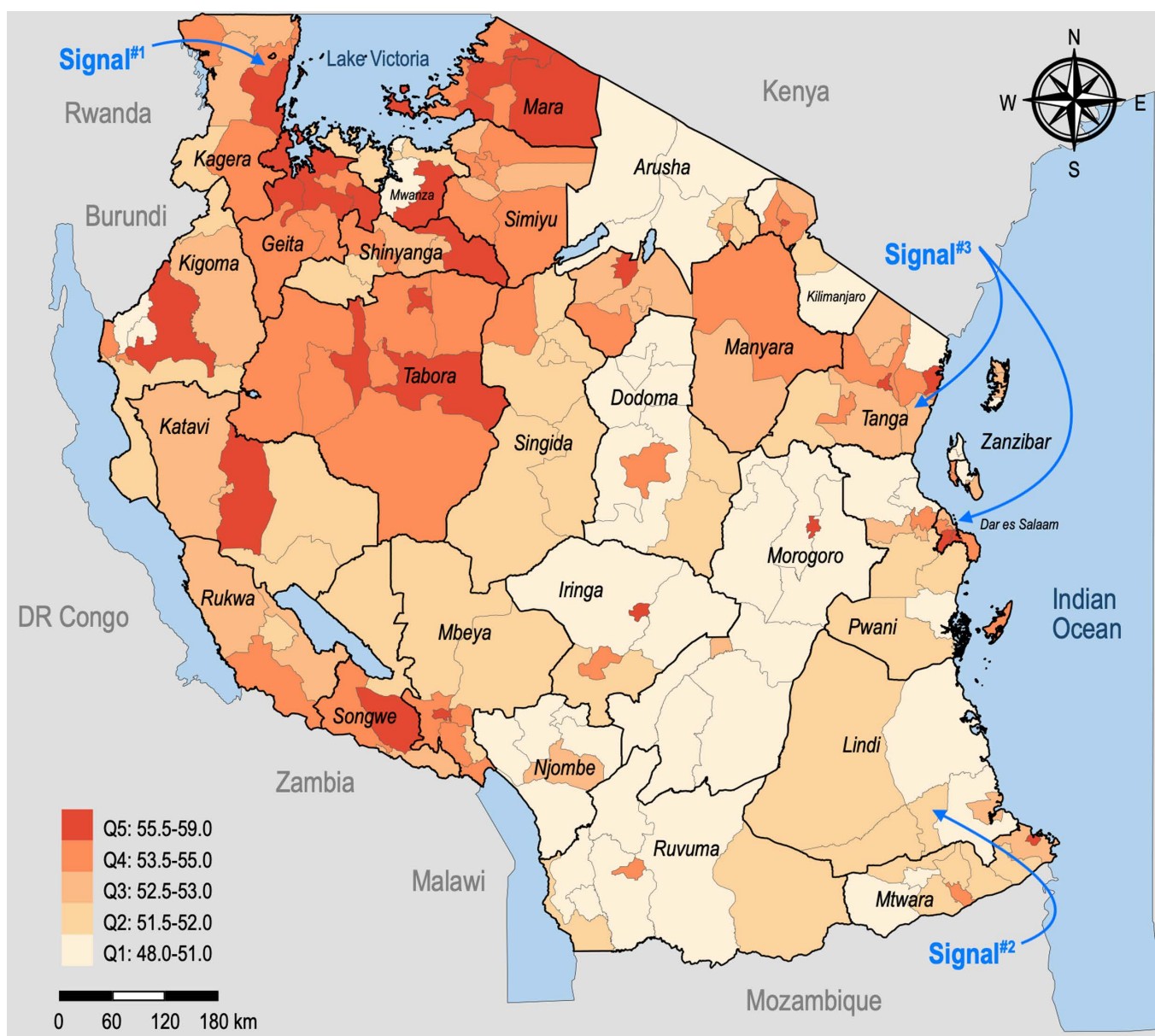

**Fig 3. Distribution of the site-specific risk at the district level in Tanzania [Shapefile source: [ 84] (this product was adapted from the National Bureau of Statistics Tanzania)].**

regions, respectively (two for short rainy season and two for detecting at least one laboratory confirmed dengue case in the last four weeks within the same region), resulting in a total aggregated public health risk score (*A+B+C*) of 66.6 in Dar es Salaam region (65-68 across all five districts) and 65.8 in Tanga region (63-70 across all 11 districts). Using the same methodology, the risk score was also calculated for all other districts where no signal was detected (corresponding to only *Risk Score B+C*). Fig 4 shows a visualization of all compartments of the integrated public health risk assessment for Tanzania at the district level, including the 20 districts most at risk of a significant public health event. The risk maps provide a comprehensive view of Tanzania's total risk score in an international context, referencing the minimum and maximum possible scores (29-132 points). Additionally, they feature a country-adjusted risk score, which uses the minimal

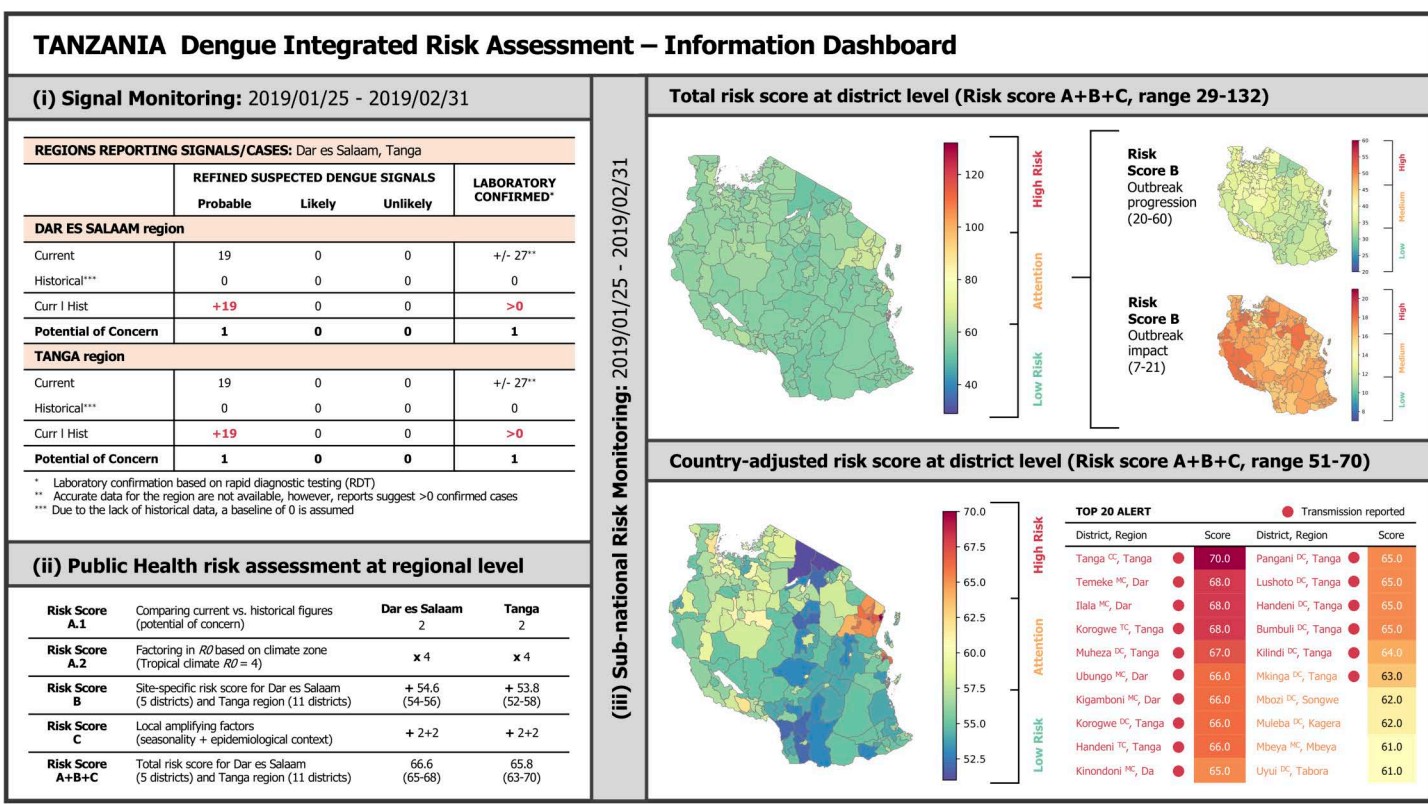

**Fig 4. Visual representation of the integrated public health risk assessment for Tanzania: (i) Signal monitoring in the reference period 2019/01/25 - 2019/02/31, (ii) Summary of the Public Health risk assessment framework for Dar es Salaam and Tanga region, (iii) Sub-national risk monitoring in the reference period 2019/01/25 - 2019/02/31, mapping both the total and country-adjusted risk score distributions for Tanzania across two distinct risk score ranges (29-132 and 51-70 points, respectively) [Shapefile source: (84) (this product was adapted from the National Bureau of Statistics Tanzania)].**

and maximal scores within Tanzania as a reference (51-70 points) to highlight national variations more clearly. Supplementary to this, the distribution of risk points along the dimensions "outbreak progression" and "outbreak impact" (only *Risk Score B*) is illustrated.

Our risk assessment indicates that the districts within the Tanga and Dar es Salaam regions are highly vulnerable to a significant dengue outbreak (level "High Risk") during the period of January 25-31, 2019. More, specifically, Tanga CC district (Tanga region) faces the highest risk, with a score of 70 points, closely followed by Ilala MC and Temeke MC (Dar es Salaam region), along with Korogwe TC and Muheza DC (Tanga region), each with a risk score of 68 points. All these districts reported active dengue transmission. Areas without active transmission but still at elevated risk (level "Attention") include, for example, Muleba DC (Kagera region), Mbozi DC (Songwe region), and Moshi MC (Kilimanjaro region), with a risk score of 62, 62 and 61 points, respectively. Conversely, districts Songea DC (Ruvuma region), Longido DC and Ngorongoro DC (Arusha region), situated mainly in the highlands, face the lowest risk level, with a score of 52, 51 and 51 points, respectively. A summary of the risk scores for each district is available in S1 Data.

## Discussion

We presented the development and evaluation of a novel integrated framework for the early detection and real-time rapid risk assessment of dengue outbreak events in Tanzania, East

Africa, using a data-driven, methodologically pluralistic, and translational approach. Tailored for settings with limited diagnostic capacity and uncertain exposure, our framework considers suspected and confirmed dengue fever cases detected through EBS (and, where applicable, complemented by routine surveillance), followed by a site-specific rapid risk characterization at subnational level to inform outbreak investigation and response. In doing so, it builds upon existing risk assessment practices, enhancing their strengths while also addressing their limitations.

In many resource-restricted settings, the early detection of infectious disease outbreaks is facilitated by EBS [12]. Thus, the incorporation and systematic processing of signals from established EBS can significantly enhance local risk assessment efforts, such as the WHO EIOS, which is also being implemented in the WHO African Region, including Tanzania [86], or other digital input sources [64]. By filtering and refining EBS signals using validated case definitions and openly available contextual data, our framework addresses the challenge of high sensitivity in dengue outbreak detection, as demonstrated by descriptive test scenarios. While our algorithm provides a rather simple refinement approach, its classification positively aligns with the actual diagnoses. To our knowledge, this is the first study to incorporate contextual intelligence at this early stage of event detection and risk assessment. However, our algorithm has not yet been tested for accuracy on a sufficient number of signals. In addition, further improvements could be achieved by integrating routine or sentinel surveillance data, as seen in countries such as Brazil or Mexico [21, 22], and by using machine learning techniques [87].

While early signal refinement allows for immediate risk profiling, both clinically (e.g., monitoring of diseased cases) and at the public health level (e.g., rapid risk assessment), laboratory confirmation is often not available until several days after identification of suspected cases due to limited resources. Therefore, the action points outlined in Table 6 need to include recommendations for relevant differential diagnosis (especially for other VHF or arboviral diseases of concern in Tanzania (see S3 Table)) and advocate for the mobilization of local resources, such as national reference laboratories in Tanzania or the East African Community mobile laboratory infrastructure [88]. In addition, any signal sensitive to suspected DHF/VHF should be reported to public health authorities, regardless of the level of refinement, as it is an immediately notifiable disease in Tanzania. This can be automated through technical interfaces with established Health Information Management Systems such as DHIS2, which is used in Tanzania [89].

We further demonstrated the feasibility of our integrated public health risk assessment framework (*Compartment III*), linked to signal identification and refinement, using the 2019 dengue outbreak as a test case. Yet, due to limited data availability, our analysis relies solely on regional data from a single event, limiting our ability to replicate our findings for other events and to adjust for potential confounders such as age, sex, and socioeconomic status. Our risk profile analysis of Tanzania revealed a significant positive association between site-specific risk scores and dengue cases at the regional level, both with and without the inclusion of temporally dynamic amplifying factors (seasonality and recent outbreak reports). Incorporating amplifying factors led to a slightly diminished effect size, an unexpected outcome given that prior research has demonstrated environmental conditions to be conducive to both mosquito survival and dengue transmission [45,47,90]. Deepening our understanding of meteorological influences and utilizing real-time data that capture local variations and climatic changes rather than using seasonal categories could lead to more precise outcomes. Our framework could be further improved by incorporating currently missing entomological data. Initial approaches to vector control are in place for Temeke district in the Dar es Salaam region [91] and could serve as important baseline information once they become routine and available for the entire

region. Additionally, incorporating ecological vector distribution models [67,92] could further enhance risk assessment and prediction, as implemented by WHO AFRO [24]. Aside from the predictive value of environmental factors in monitoring dengue transmission [93, 94], studies have shown that reported dengue cases from the previous month can aid in anticipating future outbreak hotspots [95]. Hence, further validation with sufficient data points, both epidemiological and contextual, and the use of artificial intelligence methods could refine our indicator-based framework [96], especially in the selection and weighting of indicatorsthat are currently primarily a priori informed.

As illustrated in Fig 4, translating data into actionable information at a subnational scale is critical. To achieve this, our approach systematically assigns risk scores at subnational level across different thematic compartments, and translates the total risk score into real-time alert levels (e.g., low, medium, high), aligning with established frameworks, like the Dengue Risk Mapping deployed by WHO AFRO [24], and ECDC risk assessment tool [65]. For example, our visual board indicates that Tanzania's overall risk profile is relatively low on a global scale (map "total risk score at district level"). However, country-adjusted mapping reveals that some districts may require more attention than others to respond to the situation, despite the overall low alert level. More specifically, when comparing areas with ongoing transmission, Tanga CC in the Tanga region faces a higher public health risk compared to districts in Dar es Salaam. Conversely, northeastern districts that have not yet reported cases could benefit from increased awareness (e.g., risk communication and capacity strengthening in diagnostics and clinical management), as they would be more severely affected by an outbreak than the southeast or highlands. These findings underscore the advantages of fine-scale assessments at district level compared to regional or national evaluations (e.g., WHO AFRO Dengue Risk Mapping [24]). However, action points, as proposed by WHO during previous outbreaks [63, 64], should always be developed in close collaboration with local stakeholders, while also taking into account the local infrastructure for epidemic preparedness and response [97]. In addition, a critical limitation of our framework concerns the data used to determine fine-scale risk scores, as they are gathered from various sources at different scales, with some of them being outdated or potentially biased. To address this, we conducted a data quality assessment, which should be provided to decision-makers as a supplementary resource (e.g., ECDC risk assessment tool [65]). Here, our framework could benefit from participatory refinement, e.g., by incorporating user-centered design methods [98].

Designed for rapid deployment, our integrated risk assessment framework offers methodological versatility, allowing for adaptation to different geographic contexts (e.g., other countries), scales (e.g., different subnational levels), and diseases (e.g., other hemorrhagic fevers). To achieve this, we utilized data from all United Nations member states to inform universal cut-offs for our site-specific risk scoring approach, enabling cross-country comparisons. However, we emphasize that such frameworks should not be seen as a stand-alone solution, but rather as a complementary tool to enhance EBS and routine surveillance activities, e.g., as part of local or regional early warning event management systems, and support local diagnostic capacity. In future research, we aim to investigate whether the framework would benefit from different risk contextualization (e.g., an African regional comparison rather than a global one), and to assess the feasibility of applying our approach to other countries and diseases.

## Conclusion

Developing an integrated rapid risk assessment model for dengue outbreaks could significantly enhance epidemic preparedness and response efforts in low-resource settings, especially where exposure is uncertain, such as Tanzania. Compared to existing frameworks, our

approach excels through the integration of EBS (i.e., WHO EIOS) and contextual intelligence, facilitated by systematic and automated data processing at subnational level, which allows for real-time and fine-scale risk profiling ahead of receiving laboratory confirmation. In this regard, we could demonstrate its feasibility using historical data and identify significant risk variation at the district level, which was positively associated with dengue incidence figures from 2019. Consequently, our framework can meet the information needs of a broad spectrum of stakeholders involved in the strategic planning and prioritization of interventions, covering diagnostics, clinical management, and outbreak investigation and control. With the growing availability and quality of data, coupled with the integration of artificial intelligence and user-centered design, there is significant potential for further enhancement and refinement.

## Supporting information

**S1 Text. Identification of indicators relevant to the public health risk assessment for dengue outbreaks.** COMPARTMENT I.
(DOCX)

**S2 Text. Identification of indicators relevant to the public health risk assessment for dengue outbreaks.** COMPARTMENT III.
(DOCX)

**S1 Table. Number of confirmed dengue fever cases in 2019, by month and region.**
(DOCX)

**S2 Table. (A) Summary of contextual data indicators and their characteristics, (B) Classifying "quality of evidence" and "confidence in assigning risk" informed by the ECDC's technical report "Operational tool on rapid risk assessment methodology".**
(DOCX)

**S3 Table. Viral hemorrhagic fever and arboviral diseases of public health importance in Tanzania (.docx).**
(DOCX)

**S4 Table. Glossary.**
(DOCX)

**S1 Data. Summary of the risk scores per district.**
(CSV)

## Acknowledgements

We would like to thank Kristopher Nolte, Nima Ahmady-Moghaddam, Devotha Nyambo, Jennifer Pohlmann, Walter Leal Filho, Dr. Amith Vijayakumar, Mohammed Baajour, Shinuna Gärtner, Marie Kewitz, Julia Beckhaus, Alina Nicolai, and Samantha Klein for supporting the development of the risk assessment framework through extensive review of the literature, materials, and data, and to the ESIDA network members and associates whose efforts sharpened the framework, Centre for Research, Agricultural Advancement & Teaching Excellence and Sustainability (CREATES) at NM-AIST (Nelson Mandela African Institution of Science and Technology, Arusha, Tanzania), Dr. Florian Gehre and Dr. Muna Affara (Bernhard Nocht Institute for Tropical Medicine, Arusha, Tanzania), as well as Dr. Ricardo Strauss (Bernhard Nocht Institute for Tropical Medicine, Hamburg, Germany) for their expertise in diagnostics, risk assessment, and surveillance in Tanzania and dengue-endemic regions. Finally, we extend our gratitude to the WHO EIOS team for the opportunity to utilize the EIOS system for this research.

## Author contributions

**Conceptualization:** Matthias Hans Belau, Juliane Boenecke.

**Data curation:** Matthias Hans Belau, Juliane Boenecke, Jonathan Ströbele.

**Formal analysis:** Matthias Hans Belau.

**Funding acquisition:** Mirko Himmel, Johanna Brinkel, Thomas Clemen, Wolfgang Streit, Jürgen May, Amena Almes Ahmad, Ralf Reintjes, Heiko Becher.

**Investigation:** Matthias Hans Belau, Juliane Boenecke, Jonathan Ströbele.

**Methodology:** Matthias Hans Belau, Juliane Boenecke, Jonathan Ströbele.

**Project administration:** Juliane Boenecke, Johanna Brinkel.

**Resources:** Mirko Himmel, Daria Dretvić, Ummul-Khair Mustafa, Katharina Sophia Kreppel, Elingarami Sauli, Johanna Brinkel, Ulfia Annette Clemen, Thomas Clemen, Wolfgang Streit, Jürgen May, Amena Almes Ahmad, Ralf Reintjes, Heiko Becher.

**Software:** Jonathan Ströbele, Thomas Clemen.

**Supervision:** Thomas Clemen, Jürgen May, Ralf Reintjes, Heiko Becher.

**Validation:** Matthias Hans Belau.

**Visualization:** Matthias Hans Belau, Juliane Boenecke, Jonathan Ströbele.

**Writing – original draft:** Matthias Hans Belau, Juliane Boenecke.

**Writing – review & editing:** Matthias Hans Belau, Juliane Boenecke, Jonathan Ströbele, Mirko Himmel, Daria Dretvić, Ummul-Khair Mustafa, Katharina Sophia Kreppel, Elingarami Sauli, Johanna Brinkel, Ulfia Annette Clemen, Thomas Clemen, Wolfgang Streit, Jürgen May, Amena Almes Ahmad, Ralf Reintjes, Heiko Becher.

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
