## [Decision Letter · Decision Letter 0]

20 Nov 2024

PNTD-D-24-00913Integrated Rapid Risk Assessment for Dengue Fever in Settings with Limited Diagnostic Capacity and Uncertain Exposure: Findings from a Pilot Study in TanzaniaPLOS Neglected Tropical Diseases Dear Dr. Belau, Thank you for submitting your manuscript to PLOS Neglected Tropical Diseases. After careful consideration, we feel that it has merit but does not fully meet PLOS Neglected Tropical Diseases's publication criteria as it currently stands. Therefore, we invite you to submit a revised version of the manuscript that addresses the points raised during the review process. Please submit your revised manuscript within 60 days Jan 19 2025 11:59PM. If you will need more time than this to complete your revisions, please reply to this message or contact the journal office at plosntds@plos.org. Please include the following items when submitting your revised manuscript: * A rebuttal letter that responds to each point raised by the editor and reviewer(s). You should upload this letter as a separate file labeled 'Response to Reviewers '. This file does not need to include responses to any formatting updates and technical items listed in the 'Journal Requirements' section below. * A marked-up copy of your manuscript that highlights changes made to the original version. You should upload this as a separate file labeled 'Revised Manuscript with Track Changes '. * An unmarked version of your revised paper without tracked changes. You should upload this as a separate file labeled 'Manuscript '. If you would like to make changes to your financial disclosure, competing interests statement, or data availability statement, please make these updates within the submission form at the time of resubmission. Guidelines for resubmitting your figure files are available below the reviewer comments at the end of this letter. We look forward to receiving your revised manuscript. Kind regards, Rebecca C. ChristoffersonAcademic EditorPLOS Neglected Tropical Diseases Victoria BrookesSection EditorPLOS Neglected Tropical Diseases

Shaden Kamhawi

co-Editor-in-Chief

Paul Brindley

co-Editor-in-Chief

**Journal Requirements:**

At this stage, the following Authors/Authors require contributions: Matthias Hans Belau, Juliane Boenecke, Jonathan Ströbele, Mirko Himmel, Daria Dretvić, Ummul-Khair Mustafa, Katharina Sophia Kreppel, Elingarami Sauli, Johanna Brinkel, Ulfia Annette Clemen, Thomas Clemen, Wolfgang Streit, Jürgen May, Amena Almes Ahmad, Ralf Reintjes, and Heiko Becher. Please ensure that the full contributions of each author are acknowledged in the "Add/Edit/Remove Authors" section of our submission form.

Potential Copyright Issues:

i) Figures 3, and 4. Please (a) provide a direct link to the base layer of the map (i.e., the country or region border shape) and ensure this is also included in the figure legend; and (b) provide a link to the terms of use / license information for the base layer image or shapefile. We cannot publish proprietary or copyrighted maps (e.g. Google Maps, Mapquest) and the terms of use for your map base layer must be compatible with our CC BY 4.0 license.

1) State the initials, alongside each funding source, of each author to receive each grant. For example: "This work was supported by the National Institutes of Health (####### to AM; ###### to CJ) and the National Science Foundation (###### to AM).".

**Reviewers' Comments:**Reviewer's Responses to Questions

**Key Review Criteria Required for Acceptance?**

**Methods**

-Are the objectives of the study clearly articulated with a clear testable hypothesis stated?

-Is the study design appropriate to address the stated objectives?

-Is the population clearly described and appropriate for the hypothesis being tested?

-Is the sample size sufficient to ensure adequate power to address the hypothesis being tested?

-Were correct statistical analysis used to support conclusions?

-Are there concerns about ethical or regulatory requirements being met?

Reviewer #1: The aim of the study or goal of the study is mentioned.

The objectives (small definitions of goals) in a stepwise manner can be written more clearly.

The study design is very complex and, as such, not clearly mentioned; rather, described as frameworks or compartments, which may take a lot of time or are sometimes not able to be understood by readers.

The figure 2 of context driven factors should take into account the presence of vector mosquitoes, any entomological surveillance or indicators. The other factors like previous cases and temperature alone may be not be considered sufficient to draw a framework for dengue risk assessment which depends so much on day biting vector and people's habits of storing water or recreational activities like fountains, waterpots, and the artificial collection of water in containers (container index for mosquito surveillance)...in lines 176 to 193 the investigators have themselves described the importance of vector habits and breeding patterns in dengue risk assessment, but temperature and rainy patterns alone may not decide "survival and suitability" factors in the study

I find this factor plus integrated vector management efforts and surveillance factors consideration missing in this manuscript. What about cases that are diagnosed timely and not diagnosed? They should be taken into account and mentioned clearly in simple language in a summary or in conclusions as considered suitable by authors.

The sample size sufficiency cannot be determined as existing WHO guidelines are compared against the author's work. As I understand it, there is only one author involved in data collection from one region of the continent. Unless multiple foci involving different health systems and surveillance are involved, the data or findings cannot be generalised.

The statistical analysis is described separately for different compartments. The quantity and quality of information in this article are exhausting for understanding, and sometimes figures and tables are not relatable to each other.

statistical tests used are not described; only framework for defining risk assessment is given. The basic definitions are not given. The authors should consider giving the existing terms and definitions used by researchers all over the world.

This seems like a secondary data study (infectious disease modelling) maintaining name-wise anonymity, but region- and site-wise description and risk assessment are not anonymous.

The data collection and validation guidelines can be more specific, depending on author's preferences and objectives of the study

The methods section can benefit by a table of definitions of frequently used terms in this study, eg; hazard ratio, R0, signals, risk assessment, early warning surveillance, EBS, survival signals, etc. The readers will find it easier to follow the results and conclusions with the help of table of terms and acronyms .

In Line 219, "expert discussions," please specify and give references here.

The associated details for "compartment III" are in supplementary file. A small footnote accompanying the table and explaining the columns of the table and risk level scoring is desirable in the methods.

Reviewer #2: The study is of great interest when considering the rise of arboviruses on the African continent, where many countries have an insufficient level of preparedness. Predicting dengue outbreak risk is particularly valuable to anticipate and manage future outbreaks.

The objectives are clearly stated, and the study design aligns with the hypothesis being tested.

While the methodology is generally well-conceived, I am critical of Compartment 2: Refinement of Clinical Signals Using Contextual Data. I disagree with some of the contextual factors used for refinement:

- The life history data of Aedes aegypti lacks references for development time and lifespan. The latter, which may be underestimated, requires references to be credible.

- The key factors identified for Aedes aegypti include temperature, relative humidity, and rainfall, which may be correlated in pairs.

**Results**

-Does the analysis presented match the analysis plan?

-Are the results clearly and completely presented?

-Are the figures (Tables, Images) of sufficient quality for clarity?

Reviewer #1: The whole work is comprehensive and complex. If the authors consider presenting it in more parts, more readers, researcers and care providers will be able to understand and follow their message

Authors are to congratulated on their efforts for this work and submit it in few words. I think a book or chapterwise explanation may do more justice to this project.

The actual data will need presentation. This MS describes methods and framework . More work and visual depiction of reaching results is needed to be actually incorporated in the MS. Maybe two parts of article that how framework was designed by reviewing the data and how risk assessment for Tanzania was done.

A statistical analyst may comment more on results and Chloropeth charts with review of supplementary files.

Fugires are clear.

The rsults need more clarity and comparisons

Reviewer #2: The analysis presented matches the initial plan, and the results are clearly presented. The discussion does not overlook the results. The tables and figures are sufficiently illustrative of the findings.

**Conclusions**

-Are the conclusions supported by the data presented?

-Are the limitations of analysis clearly described?

-Do the authors discuss how these data can be helpful to advance our understanding of the topic under study?

-Is public health relevance addressed?

Reviewer #1: The methods is novel and designed with appropriate reviews and integration but how it was utilised in field and actual situation is not clear from results and conclusions.

Reviewer #2: The study addresses the important topic of dengue risk prediction and provides a framework for risk assessment in Tanzania that could be applied to other countries.

**Editorial and Data Presentation Modifications?**

Reviewer #1: I would like to express my gratitude to the Journals Editorial Team for giving me this opportunity to review this article.

The amount of work done and statistical modelling and concepts are novel, though somewhat larger in quantity and quality. Average readers and researchers can find this article difficult to understand and apply in their settings.

The content has to be read again and again for understanding the concept.

Morever, how these new risk assessment strategies may benefit community or health program implementation needs more context and clarity.

I have another query in my mind: Usually in any situation, we recommend strengthening diagnosis and surveillance; then should we not focus more on strategies to improve surveillance rather than preparing newer frameworks based on assumptions and definitions and existing criteria?

Reviewer #2: (No Response)

**Summary and General Comments**

Reviewer #1: The article is written to suggest a new integrated framework for dengue risk assessment in context-specific settings of Tanzania in east Africa.

The authors state that existing WHO Epidemic Intelligence from Open Source Systems (EMIOS), e.g., WHO EWRS for dengue, requires continuous surveillance and laboratory services networks. These ideal situations of diagnosis and surveillance are not present in Tanznaia and other LMIC settings, and there is a need for event-based surveillance (EBS).

1. The title says- Limited Diagnostic Capacity and Uncertain Exposure: Findings from a Pilot Study. What were criteria to decide that the diagnostic capacities were limited and exposure was uncertain?

2. The work is comprehensive but very difficult to understand. Is this compilation of several articles for determing risk assessment or analysis of cases from Tanzania or as stated in introduction report of recent outbreak.

3. Real- world data—WHO and Tanzania. (how is that collected and compared) and in what manner integrated with existing EWS; should be written clearly.

4. In what ways is the discussed- "integrated surveillance assessment system" NOVEL from existing approaches

Reviewer #2: The study is of great interest when considering the rise of arboviruses on the African continent. By developing a framework of dengue risk prediiction for Tanzania, the authors prvide tools to develop moodels for other African countries where dengue is on the rise.

PLOS authors have the option to publish the peer review history of their article (what does this mean? ). If published, this will include your full peer review and any attached files.

**Do you want your identity to be public for this peer review?** For information about this choice, including consent withdrawal, please see our Privacy Policy .

Reviewer #1: No

Reviewer #2: **Yes: ** Athanase Badolo

**Figure resubmission:**

**Reproducibility:** To enhance the reproducibility of your results, we recommend that authors of applicable studies deposit laboratory protocols in protocols.io, where a protocol can be assigned its own identifier (DOI) such that it can be cited independently in the future. Additionally, PLOS ONE offers an option to publish peer-reviewed clinical study protocols. Read more information on sharing protocols at https://plos.org/protocols?utm_medium=editorial-email&utm_source=authorletters&utm_campaign=protocols

---

## [Editor Report · Decision Letter 1]

17 Feb 2025

PNTD-D-24-00913R1Integrated Rapid Risk Assessment for Dengue Fever in Settings with Limited Diagnostic Capacity and Uncertain Exposure: Findings from a Pilot Study in TanzaniaPLOS Neglected Tropical Diseases  Dear Dr. Belau, Thank you for submitting your manuscript to PLOS Neglected Tropical Diseases. After careful consideration, we feel that it has merit but does not fully meet PLOS Neglected Tropical Diseases's publication criteria as it currently stands. Therefore, we invite you to submit a revised version of the manuscript that addresses the points raised during the review process. Please submit your revised manuscript within 30 days Mar 19 2025 11:59PM. If you will need more time than this to complete your revisions, please reply to this message or contact the journal office at plosntds@plos.org. Please include the following items when submitting your revised manuscript: * A rebuttal letter that responds to each point raised by the editor and reviewer(s). You should upload this letter as a separate file labeled 'Response to Reviewers '. This file does not need to include responses to any formatting updates and technical items listed in the 'Journal Requirements' section below. * A marked-up copy of your manuscript that highlights changes made to the original version. You should upload this as a separate file labeled 'Revised Manuscript with Track Changes '. * An unmarked version of your revised paper without tracked changes. You should upload this as a separate file labeled 'Manuscript '. If you would like to make changes to your financial disclosure, competing interests statement, or data availability statement, please make these updates within the submission form at the time of resubmission. Guidelines for resubmitting your figure files are available below the reviewer comments at the end of this letter. We look forward to receiving your revised manuscript. Kind regards, Rebecca C. ChristoffersonAcademic EditorPLOS Neglected Tropical Diseases Victoria BrookesSection EditorPLOS Neglected Tropical Diseases

Shaden Kamhawi

co-Editor-in-Chief

Paul Brindley

co-Editor-in-Chief

**Additional Editor Comments:** The title could be changed to better reflect... something like: "Development of a methodological framework for ...." would be a better representation rather than "findings from a pilot study" which infers a more data research study. This study is a framework validation, as presented.

In the discussion, there could be more discussion on how this framework fill existing gaps and how it could be integrated into existing systems - such as MoH in Tanzania **Journal Requirements:**

Please ensure that the funders and grant numbers match between the Financial Disclosure field and the Funding Information tab in your submission form. Note that the funders must be provided in the same order in both places as well.

State the initials, alongside each funding source, of each author to receive each grant. For example: "This work was supported by the National Institutes of Health (####### to AM; ###### to CJ) and the National Science Foundation (###### to AM).".

**Reviewers' comments:** **Figure resubmission:** While revising your submission, please upload your figure files to the Preflight Analysis and Conversion Engine (PACE) digital diagnostic tool, https://pacev2.apexcovantage.com/. PACE helps ensure that figures meet PLOS requirements. To use PACE, you must first register as a user. Registration is free. Then, login and navigate to the UPLOAD tab, where you will find detailed instructions on how to use the tool. If you encounter any issues or have any questions when using PACE, please email PLOS at figures@plos.org. Please note that Supporting Information files do not need this step. If there are other versions of figure files still present in your submission file inventory at resubmission, please replace them with the PACE-processed versions. **Reproducibility:** To enhance the reproducibility of your results, we recommend that authors of applicable studies deposit laboratory protocols in protocols.io, where a protocol can be assigned its own identifier (DOI) such that it can be cited independently in the future. Additionally, PLOS ONE offers an option to publish peer-reviewed clinical study protocols. Read more information on sharing protocols at https://plos.org/protocols?utm_medium=editorial-email&utm_source=authorletters&utm_campaign=protocols

---

## [Editor Report · Decision Letter 2]

25 Feb 2025

Dear Dr. Belau,

We are pleased to inform you that your manuscript 'Integrated Rapid Risk Assessment for Dengue Fever in Settings with Limited Diagnostic Capacity and Uncertain Exposure: Development of a Methodological Framework for Tanzania' has been provisionally accepted for publication in PLOS Neglected Tropical Diseases.

Best regards,

Rebecca C. Christofferson

Academic Editor

Victoria Brookes

Section Editor

Shaden Kamhawi

co-Editor-in-Chief

Paul Brindley

co-Editor-in-Chief

---

## [Editor Report · Acceptance letter]

Dear Dr. Belau,

We are delighted to inform you that your manuscript, "Integrated Rapid Risk Assessment for Dengue Fever in Settings with Limited Diagnostic Capacity and Uncertain Exposure: Development of a Methodological Framework for Tanzania," has been formally accepted for publication in PLOS Neglected Tropical Diseases.

Best regards,

Shaden Kamhawi

co-Editor-in-Chief

Paul Brindley

co-Editor-in-Chief
